# ENSO-driven hypersedimentation in the Poechos reservoir, northern Peru

Anthony Foucher[1], Sergio Morera[2,3,4], Michael Sanchez[2], Jhon Orrillo[2] & Olivier Evrard[1]

[1] Laboratoire des Sciences du Climat et de l'Environnement (LSCE-IPSL), UMR 8212 (CEA/CNRS/UVSQ), Université Paris-Saclay, 91191 Gif-sur-Yvette Cedex, France
[2] Instituto Geofísico del Perú, Calle Badajoz, 169, Lima, Peru
[3] Universidad Nacional Agraria La Molina, Av. La Molina, s/n, Lima, Peru
[4] Universidad Nacional Autónoma de Huanta, Jr. Manco Cápac, 497 Ayacucho, Peru

*Correspondence to*: Anthony Foucher (Anthony.foucher@lsce.ipsl.fr)

**Abstract.** Although Extreme El Niño Events (EENE) have always impacted hydrological anomalies and sediment transport in South America, their intensification by global warming and their association with changes in human activities and land covers after humid periods may lead to the acceleration of sediment transfers in river systems and dam reservoirs. This situation may threaten soil and water resources in arid and semiarid regions highly dependent on water originating from large dams. In this study, we investigated the sediment sequence accumulated in the Poechos reservoir (northern Peru) and provided a retrospective reconstruction of the interactions of El Niño-Southern Oscillation (ENSO), agricultural practices and vegetation cover changes on sediment dynamics (1978-2019). To this end, a sediment core was dated and characterized by physical and chemical analyses (e.g. scanner tomography, X-ray fluorescence, particle size analysis) for estimating the evolution of sedimentation rates and changes in sediment sources during the last five decades.

Sediment tracing results indicated the occurrence of changes in sediment sources associated with positive and negative phases of the Eastern Pacific Index with a greater contribution of the lowland dry forest area in comparison to that of the Andean area to sediment during the El Niño Events, (mean contribution of 76%, up to 90% during the Coastal El Niño Events (CENE) of 2016-2017). This source contribution was mostly controlled by the stationary rainfall occurring during the Extreme El Niño Events (EENE) in the lowland dry forest area characterized by a low vegetation cover. Overall, after an extreme phase of ENSO, like after the EENE 1982-1983, the normal discharges and persistent sediment supplies from the middle and upper catchment parts led to river aggradation and the storage of substantial amounts of sediment in alluvial plains. In the absence of significant EENE event between 1983 and 1997, the large volume of sediment stored in the alluvial plains was exported by the EENE 1997-1998 resulting into an increase in sedimentation rate by 140 % after 1997 with a significant aggradation of the deltaic zone of the reservoir. In addition to the impact of extreme climate events on sediment dynamics, the development of agriculture along the riverine system after an extreme phase of ENSO increased the availability of sediments in the main channel of the rivers, easily transported by the next EENE. This study suggests that intensification of human activities associated with a higher frequency of extreme rainfall events amplified the quantity of sediment transported by the river system, which will significantly decrease the lifespan of the reservoir essential to meet freshwater demands of the farmers and the populations living in this arid and semiarid region.


**Keywords:** Sediment core, Extreme El Niño events, Sediment tracing, Land use change, Water resources, Agriculture expansion

## 1 Introduction

In many regions of the world, the conversion of natural ecosystems into agricultural land was observed during the last several
decades to meet the local and global demand for food and other commodities of a growing population (Winkler et al., 2021). South-America is one of the most affected continents by this recent phase of agricultural expansion and land cover change (Foucher et al., 2023; Song et al., 2021). Zalles et al. (2021) estimated that 60% of this continent's land surface was impacted by human activities, specially through land use conversion and natural land cover modifications from 1985 to 2018 (e.g. expansion of soybean at the expense of natural habitats including native grasslands and primary forests).
In addition, climate projections indicated an expected change in the distribution, frequency and intensity of rainfall with increasing periods of drought and extreme precipitation in this part of the world. As an example, rainfall in southern Chile and southwestern Argentina was shown to have been decreasing since 1960 while at the same time it has been increasing in northwestern Peru and Ecuador (Barros et al., 2014). The occurrence of extreme rainfall events in catchments impacted by land cover changes induced by natural (e.g. vegetation blooms after rainfall) or anthropogenic factors were shown to accelerate
soil erosion and induced deleterious consequences downstream including muddy floods (e.g. McEntire and Fuller, 2002; Morera et al., 2017; Strozzi et al., 2018), or an increased transfer of particle-bound contaminants and the siltation of water bodies (e.g. Custodio et al., 2022). By analysing data on sediment load available from gauging stations, reservoir sedimentation and water turbidity over a 30-yr period, Rosas et al. (2023) have demonstrated high variations in suspended sediment yield in the north of the western Peruvian Andes ($3\circ - 6\circ$ S) varying from 12 to 2 330 t km$^{-2}$ yr$^{-1}$ similarly to previous studies for the
northern Andes (e.g. sediment yields of 900 to 1150 t km$^{-2}$ yr$^{-1}$ reported for rivers draining the western flank of the Ecuadorian Andes (Vanacker et al., 2015) and suspended sediment yield ranging between 16 t km$^{-2}$ yr$^{-1}$ and > 15 000 t km$^{-2}$ yr$^{-1}$ for small degraded catchments in southern Ecuador (Molina et al., 2008; Vanacker et al., 2007).

The South-American continent is highly dependent on water resources stored in dams for irrigation, hydroelectric power production (60% of electricity demand supplied by hydroelectric power generation according to the International Energy
Agency) and human commodities (Paredes-Beltran et al., 2021).

A recent publication has demonstrated the vulnerability of Andean hydroelectric reservoirs against future climate change (Rosas et al., 2020). The siltation of these reservoirs with sediment that may be contaminated with pesticides or other pollutants may impair the quality and quantity of the water resources available, which may lead to multiple deleterious economic, environmental and health consequences (de Campos et al., 2020), in one of the most densely populated mountain areas on
Earth.

Peru is among the countries that are the most strongly impacted in South America by extreme rainfall events and widespread land cover changes (e.g. deforestation of both humid and dry forests, expansion of agriculture), (e.g. Barboza et al., 2022). The climate of the north Peru is highly controlled and influenced by the El Niño-Southern Oscillation (ENSO) involving changes in the temperature of seawater in the central and eastern tropical parts of the Pacific Ocean following a cycle lasting for 3 to 7

years. This oscillation directly affects rainfall distribution in Peru as in other regions of the world. Geng et al. (2022) indicate that the ENSO sea surface temperatures (SSTs) exhibit diverse anomaly centres, ranging from the equatorial eastern-Pacific (EP) to the equatorial central-Pacific (CP), referred to as EP-ENSO and CP-ENSO regimes, respectively. The EP-ENSO regime is characterised by stronger warm-than-cold SST anomalies, whereas the CP-ENSO regime features larger cold-than-warm SST anomalies. As extreme phases of ENSO can correspond to either cold (La Niña) or warm (El Niño) regimes, the

impacts on land may be completely different. The very strong EP-ENSO or El Niño generate the Extreme El Niño Events (EENE), which produce catastrophic floods in lower and intermediate sections of the Chira-Piura catchment whereas La Niña CP-ENSO generates intense drought periods.

During the recent history (post-1950), several EENE (e.g. 1982-1983, 1997-1998) and Coastal El Niño Events (CENE, e.g. 2016-2017) were observed with deleterious environmental consequences in Peru (e.g. floods, landslides) and human losses.

These EENE were generated by intense warming in the far-eastern Pacific and were usually linked with disproportionately large rainfall anomalies in the dry western coast of South America (e.g. Woodman, 1999). These EENE correspond to a different dynamical regime from the rest of ENSO due to the non-linear activation of deep convection in the cold eastern Pacific (Takahashi and Dewitte, 2016). Contrary to EENE, the CENE are not predictable. The role of EENE on sediment transport was demonstrated at the scale of the western Andes by Morera et al. (2017) indicating that the suspended sediment

yield increase by 3–60 times during EENE compared to normal years. According to these authors, this effect can be explained by a sharp increase in river water discharge due to high precipitation rates and transport capacity during these events.

Even if the damage associated with these major events is difficult to evaluate (e.g. floods, muddy floods, landslides), the EENE recorded in 1997-1998 has induced, for example, more than US$100 million of damage (CTAR, 1998). To adapt to these extreme climatic conditions and alternating dry and wet periods on the Peruvian coast (arid and semiarid regions), ancient

civilisations (1100 to 500 BC) were using floodwater farming for irrigation and flood mitigation relying on channel management (Caramanica et al., 2020). Nowadays, Farmers are still using water from reservoirs for irrigation to develop agriculture and also implemented large-scale management measures to expand agriculture along the riverine system. This management phase generated extensive soil disturbance, which may exacerbate the transport of sediment to lower river sections during the EENE (Marin, 2020).

Several authors have investigated the sediment transport along the western Peruvian Andes using gauging stations (e.g. Morera et al., 2017). They described with a high temporal resolution sediment dynamics during short time periods. Nevertheless, these time series are generally discontinuous (e.g. destruction of the gauging station during extreme events) and sediment properties were rarely characterized.

To overcome these limitations, sediment accumulated in dam reservoirs may provide a powerful alternative for continuously

reconstructing sediment properties and dynamics in this region (Rosas et al., 2023). Although limnogeological studies were widely used for paleoclimatic and paleoenvironmental reconstructions in the Northern Hemisphere (e.g. Alps, Great Lakes of USA), much less studies based on the analysis of lacustrine archives focusing on past decades (after 1950) were conducted in the Southern-Hemisphere in general, and in Peru particular (Foucher et al., 2021). This lack of studies can likely be explained by the low number of freshwater bodies in this region, especially on the western edge of the Andes. Most of the existing

limnological studies focusing on ENSO were consequently focused on marine sediment cores (Hendy et al., 2015) or on Holocene paleo reconstruction (e.g. Wells, 1990). The relative lack of studies can also be explained by the complexity to date these recent sediment archives (>1950) using radionuclides in this this low-latitude area with limited thermonuclear bomb fallout (Chaboche et al., 2021).

Accordingly, the current research investigated, as a representative study case, the sediment accumulated in the Poechos

reservoir (located on the west coast of northern Peru) for retrospectively reconstructing the impact of extreme phases of the ENSO, land cover changes after humid periods and agriculture expansion along the riverine system on sediment dynamics (1978-2019). To this end, a sediment core was dated and characterized by multiple physical and chemical analyses for (1) estimating the evolution of sedimentation rates and (2) identifying the sources of sediment participating to dam siltation. The main goal of this study is to improve our understanding of the sediment dynamics in one of the most densely populated

mountain regions of the world where the sustainability of soil and water resources is of major concern. The identification of sediment sources and their relationship with sediment rates can help improving upstream watershed management and soil conservation programs to limit the adverse effects of accelerated soil erosion on the reservoir siltation whereas both consequences of climate change and the anthropogenic pressures are projected to increase in the coming decades

## 2 Site and Materials & methods

### 2.1 Study site

The Catamayo-Chira catchment (13,565 km²) is located in the northern part of Peru at the boundary with Ecuador (46 and 54% of the catchment surface area are located in Peru and Ecuador, respectively), (Fig. 1). This catchment is characterised by a contrasted relief with a lowland area in the western part and the foothills of the Andean mountains in the eastern part (the west-east gradient ranges from 81 m a.s.l at the catchment outlet to 3958 m a.s.l in the eastern part of the basin). Geology of the

catchment is mainly dominated by the occurrence of intrusive rocks from Neocene and Palaeocene (volcanic, granitoid formations) in the Andes (e.g. *Formacion Volcanico Porculla, Volcanico Llama*) and by Upper Cretaceous sedimentary and Volcanic formations in the lowland area (Vílchez et al., 2006). Two mains ecoregions are found in this basin: The Tumbes-Piura dry forests in the lowland section (coastal dessert area) and the eastern cordillera forest in the upper Andean part. According to the simplified land covers map generated in this study, the dominant land cover of the catchment corresponds to

agricultural land with 35.2% (4775 km²) of the catchment area (Fig. 1), it is distributed across the entire catchment, although

with a greater presence in the middle and upper parts, where rainfall is more frequent, contrarily to the situation in the lower part, where agriculture is restricted to areas with access to water from rivers or the Poechos reservoir. Numerous different crops are cultivated in the catchment (e.g. rice, cotton, corn, beans, cassava, sweet potatoes, potatoes, coffee, cocoa) (Otivo, 2010). Forty percent of the Economically Active Population of the catchment is linked to agriculture and to extractive activities in the forests (ANA, 2015). The dry forest covers 30.5% (4142 km$^2$) of the catchment and is mainly found in the lower part of the catchment. This type of land cover is particularly adapted to the regime and amount of rainfall found in this area, and the dry forest is being mainly degraded by the use of wood for firewood, illegal logging and change of land use associated with agriculture expansion (Otivo M. et al., 2014). According to MINAM (2016), on the coast, agriculture with technified irrigation has been reclaiming land from the dry carob forests, and every day a greater number of wooded areas are used for livestock and beekeeping. The herbaceous and/or shrubby vegetation covers 22.4% of the catchment (30322 km$^2$), mostly in the middle and upper parts of the catchment, in areas where crop establishment is difficult. The forest covers 6.6% of the surface (896 km$^2$), including natural and non-natural plantations. From 1986 to 1996, Oñate-Valdivieso and Bosque Sendra (2010) have reported minor changes in land cover dynamics in this catchment. The majority of changes were identified in the lower part of the basin where crop areas replace on dry forest biomes. This trend also occurs in the middle and high parts of the basin where natural vegetation is converted to grassland and crops for agricultural purposes.

This catchment drains into the strategic Poechos reservoir (occupying a surface area of 115 km² during the humid period). This water body located at the catchment outlet was built in 1976 to provide water for local agriculture (approximately 35,000 ha are irrigated), electric power production and to prevent flood risk. This reservoir is supplied by the Chira river (approximately 750 km of river network across the catchment). This reservoir suffers from a continuous reduction of its capacity due to the excessive supply of sediment during the extreme humid phases of the ENSO from the sources located in the vicinity and upstream of the reservoir (e.g. Tote et al., 2011). The greatest accumulation has occurred during EENE years 1983, 1998 and CENE 2017 when sediment production and transport were found to be greater than during years of average rainfall (3 to 60 times during EENE compared to normal years (Morera et al., 2017)). A bathymetric study showed that the storage loss in the active volume of the reservoir between 1976 and 1998 was 37.9%, and between 1976 and 2017 the storage loss reached 58.4% (Marin, 2020).

## 2.2 Materials & methods

### 2.2.1 Sampling

The sediment core sampling in the Poechos reservoir was carried out following a meticulous technical approach, considering various variables to ensure the representativeness of the obtained data. Firstly, an accumulated sediment zone was selected in the reservoir based on a comprehensive analysis of the bathymetry data obtained in the framework of the Chira project. This allowed the identification of areas with higher sediment deposition and the determination of zones that are more prone to sediment accumulation by considering the water flow velocity. To capture spatial variability and obtain a representative

sample, around 10 sediment cores were extracted at different points in the reservoir during two consecutive years, specifically during the dry season. These samples were collected in the upper part of the reservoir, taking advantage of its temporary emptiness during certain periods of the year, facilitating the retrieval of a 4.5-meter sediment core using a COBRA TT vibracorer equipped with 90 mm PVC liners available at the Geophysical Institute of Peru (Fig. 1). This strategic location also enabled the recording of sedimentary inputs associated with significant climatic events near the deltaic area. Upon completion of the sampling process, the sediment core, identified by the IGSN number 10.58052/IEFOU0009 and located at coordinates -80.462745; -4.570902, was obtained, providing an accurate representation of the sediment distribution in the Poechos reservoir. These multiple precautions ensure that the obtained sediment core provides a reliable tool for understanding sediment dynamics in the reservoir and supporting comprehensive analysis and future assessments. This sedimentary sequence was sectioned into six sections for transportation with a length ranging between 51 and 61 cm.

The potential sources of sediment (n=17 composite samples composed of 5 subsamples) were sampled in 2019 for tracing the origin of sediment accumulated in the reservoir. Two main sources were targeted base on the main geology units that correspond to different land cover classes of this area, i.e. the Andean area (covered by forest, agriculture and herbaceous formations) and the dry forest lowland area (occupied by dry forest), located in the upper and lower parts of the catchment, respectively (Fig. 1). Source sampling was performed using a metallic trowel and consisted in collecting the uppermost layer of the soil (≈ upper two centimeters). No channel bank samples were collected because the objective of the tracing method implemented in this study was not to distinguish channel banks from topsoils, but instead to distinguish overall sediment contributions from either upper or lower catchment parts, characterised by different lithologies.

### 2.2.2 Laboratory analyses

Sediment core sections were analyzed with an Avaatech X-Ray Fluorescence core scanner (XRF) available at the Laboratoire des Sciences du Climat et de l'Environnement (Gif-sur-Yvette, France) to obtain high resolution (0.5 cm) and semi-quantitative (cps) values of chemical elements along the core. These data were used for characterizing potential changes in sediment sources throughout time associated with land cover and climate changes across the basin. Titanium (Ti), potassium (K), strontium (Sr), rubidium (Rb) and the titanium/calcium ratio (Ca/Ti) were used for identifying potential changes in detrital material contributions along the sequence (Croudace and Rothwell, 2015).

Particle size analysis was performed using a laser grain sizer Malvern Mastersizer 3000 allowing to measure the grain size distribution between 10 nm and 3.5 mm. Particle size was measured on the sandy layers identified along the core (n=19) and on randomly selected samples along the sequence (n=10). Grain size parameters such as d10, d50, d90 and raw data were extracted for characterizing these properties.

Non-calibrated sediment density was recorded every 0.6 mm along the sediment sequence using Computer Tomography scanner (CT-Scan) images obtained using the equipment (GE Discovery CT750 HD) available at the DOSEO platform (Université Paris-Saclay, CEA, List). Image reconstruction and relative density values (CT-number) were extracted from the

scanner images using the free software ImageJ (Schneider et al., 2012) following the procedure described in Foucher et al. (2020).

Energy dispersive X-ray fluorescence (ED-XRF) measurements (Epsilon 3, Malvern Panalytical) were conducted on selected samples along the sediment cores to obtain concentrations in major elements and calibrate the relative concentrations obtained with the semi-continuous XRF analyses. ED-XRF measurements were conducted on 0.5g of crushed sediment. Calibration was performed using the free software package XELERATE (http://www.ascar.nl/). In addition to the analyses performed on the sediment core, ED-XRF measurements were also conducted on the 17 source samples.

The R package FingerPro was used for estimating the sediment contributions supplied by the potential sources (dry forest vs Andean area) in the sediment accumulated in the reservoir based on the ED-XRF and calibrated XRF records from the sediment sequence. This model un-mixed the sediment sources after a three-step statistical tracer selection procedure described in Lizaga et al. (2020).

### 2.2.3 Sediment core dating

Core chronology was established by fitting the E (Eastern Pacific) index with the CT-number extracted from the computer tomography scanner imageries. The E index (positive and negative values correspond to warm and cold events, respectively) was defined by Takahashi et al. (2011) from monthly Sea Surface Temperature (SST) anomalies to describe the extreme warm events along the north Peruvian coast (this index is available at: http://190.187.237.251/datos/ecindex_ersstv5.txt). Correlation between these temporal series was manually performed using the updated version of the free software Analyseries (Paillard et al., 1996), Qanalyseries (Kotov and Paelike, 2018).

Age-depth model validation was performed by comparing the sedimentation rate (SR expressed in $cm.yr^{-1}$) reconstructed by Qanalyseries with the annual volume of sediment accumulated in the reservoir as an independent control, estimated by annual bathymetric surveys (Marin, 2020) and by comparing the age at the base of the core with the age of reservoir impoundment (paleo-soil was reached during coring operation). In addition, gamma spectrometry measurements were obtained using HPGe detectors (Canberra/Ortec) available at the Laboratoire des Sciences du Climat et de l'Environnement. Short-lived radionuclides (caesium-137 ($^{137}Cs$) and excess of lead-210 ($^{210}Pb_{ex}$)) were measured in 12 samples of dry sediment ($\approx$10g) collected along the sedimentary sequence (approximatively every 40 cm). These results were not used for the establishment of the age model as discussed in the following sections.

### 2.2.4 Land cover map

To classify land cover (Fig. 1), information was collected in the field between 2018 and 2019, which enabled us to characterize the categories of land cover to consider, and obtain reference samples for the classification and validation. The multivariable digital classification was used, based on the protocol of the Ministry of the Environment of Peru (MINAM, 2014) and the Random Forest model was used as an object-oriented multivariable classifier. The classification process was executed using the R software, specifically applying the Random Forest model from the RandomForest package.

The classification process was carried out in 4 stages, (1) the selection of Landsat 8 OLI images in the dry season of 2016 and digital elevation model (DEM) NASA SRTM 30m, (2) radiometric calibration, atmospheric correction and topographic correction of the multispectral images were carried out. (3) the segmentation and calculation of predictor variables for the training of the model was carried out, such as: slope, normalized differential vegetation index (NDVI), (Rouse et al., 1973), Tasseled Cap spectral transformation (Baig et al., 2014) translated into indices of brightness, greenness and humidity. (4) the classification was performed using the Random Forest model (Breiman and Cutler, 2012) using the multispectral variables, the predictor variables (brightness, greenness, humidity, NDVI, DEM and slope) and reference samples, finally the accuracy of the model was analyzed followed by a vector edit of the misclassified objects.

## 3 Results

### 3.1 Core chronology

Correlation between E index and relative density extracted from the sediment core (r² of 0.45) was used to provide the chronology of the 19CO3 core. The age-model covered the period ranging between 1978 and 2019, with the lower layer corresponding approximately to the impoundment of the reservoir and the upper layer corresponding to the year of the core sampling, (Fig. 2). No $^{210}Pb_{xs}$ decrease was observed along this sediment core. Values of Lead-210 were ranging between 10 to 6 Bq kg$^{-1}$, respectively from the top to the bottom of the sequences without any significant decreasing relationship with depth. The artificial $^{137}Cs$ radionuclide was not detected at all along this sediment core. Consequently, both radionuclides could not be used for the age-model validation.

The highest SR of the 19CO3 core were observed between 1982-1983, 1997-1998, 2008-2009 and 2016-2017 with average SR values of 10, 50, 23 and 31 cm yr$^{-1}$ respectively (Fig. 2). These periods correspond to the historical EENE, ENE (El Niño Events) and CENE The identification of these historical EENE, CENE and ENE in the sediment core was used to validate the core chronology. In addition to the increases in SR associated with these events, a general shift in SR was observed in this sequence. An average SR of 5.7 cm yr$^{-1}$ (SD 2.5 cm yr$^{-1}$) was observed for the 1978-1996 period, whereas a higher mean SR of 13.7 cm yr$^{-1}$ (SD 9 cm yr$^{-1}$) was found between 1997 and 2019, corresponding to a 140% increase after 1997.

### 3.2 Lithology

The 19CO3 core was composed of a succession of coarser layers (d50 ranging between 24 and 287 μm) and of finer sediment layers (d50 ≈ 10 μm) in the upper part of the sequence (respectively between 1996 and 2019). In the lower part corresponding to material deposited between 1978 and 1996, fine and homogenous sediment was found (d50 ≈ 9 μm). A total of 19 coarser layers were visually observed in this sequence (Fig. 3 & 4). The main coarser layers were observed in 1997, 2008, 2011 and 2016 with a corresponding thickness of 7.5, 12, 7.5 and 25.5 cm. No such layer was observed before 1996. Thickness, grain size properties and age of these layers are detailed in Table 1.

Changes in geochemical contents such as those shown by Ca/Ti and Sr values highlighted the occurrence of a significant positive trend (Mann Kendall test, *p*-value <0.05) during the 1978-2019 period whereas Ti and K showed a negative trend (*p*-value <0.05). No statistical trend was observed for Rb (*p*-value of 0.3). The highest Ca/Ti and Sr values were recorded between 2007 and 2012 as well as during the 2015-2017 period. Positive fluctuations in Ca/Ti and Sr contents were mainly observed during the positive E index periods whereas the Ti behaviour showed the opposite trend with a decrease in Ti values during the positive E index periods (Fig. 3). The main changes in Ti contents were observed during the 1982-1984, 1991-1992, 1996-1998 and 2016-2017 periods corresponding to EENE, ENE and CENE as well between 2007 and 2012 without any association with EENE or ENE. Finally, the K and Rb contents showed very similar fluctuations ($r^2$ of 0.73 between both elements) (Fig. 4).

### 3.3 Sediment sources

Two tracing properties (K and Rb) were selected by the FingerPro statistical procedure for un-mixing the sediment source contributions along the 19C03 sequence (Fig. 5). K content ranged between 0.6 and 2.25% in the soil samples. The highest K contents were measured in the Andean potential sediment sources with an average value of 1.5% (SD 0.4%, ranging between 0.9 and 2.25%) whereas a lower average value of 0.9% (SD 0.2%, ranging between 0.6 and 1.2%) was recorded in the lowland dry forest source. A similar trend was observed for the Rb content with the highest values measured in the upstream sources (0.07%, SD 0.01%, ranging between 0.06 and 0.1%) and the lowest values in the downstream source (0.05%, SD 0.01%, ranging between 0.03 and 0.06%).

Calibrated K values from the sediment core ($r^2$ of 0.92 between XRF and ED-XRF calibration) varied between 0.7 and 1.8% with an average value of 1.1% (SD 0.15%). The average calibrated Rb value ($r^2$ of 0.8 between XRF and ED-XRF calibration) reached 0.05% (SD 0.01%) with values ranging between 0.03 and 0.12% (Fig. 5).

FingerPro results showed that 31% (SD 22%) of the sediment accumulated in the core was originating from the Andean source, while 69% (SD 22%) of the material was supplied by the lowland dry forest source (Fig. 6). According to the Mann-Kendall test, the sediment contribution of the dry forest followed a positive trend between 1978 and 2019 (*p*-value <0.05). On average, the dry forest source contributed 76% (SD 20%) of the sediment accumulated in the reservoir during the positive E index phases. During the negative E index periods, Andean source contribution increased and supplied 40% (SD 22%) of the material accumulated in the reservoir.

The contribution of the dry forest sources during the EENE1982-1983, 1997-1998 and CENE2016-2017 reached 94% (SD 4%), 65% (SD 25%) and 90% (SD 9%), respectively.

### 4 Discussion

Fallout radionuclide measurements could not be used for the establishment of the core chronology due to the low activities of $^{210}$Pb$_{ex}$ and the absence of $^{137}$Cs in the core sediment. Past studies had demonstrated the low $^{137}$Cs activities found in soils and

sediment in this low-latitude region of South-America (Chaboche et al., 2021) whereas the $^{210}Pb_{ex}$ was supposed to be present in the upper part of the soil (≈5cm) before decreasing with depth. The very low lead-210 activities detected in the sediment core and sediment sources can provide information on the intensity of erosion processes in this region by mobilizing subsoil sources (channel bank, gullies) depleted in $^{210}Pb_{ex}$, and consequently highly impacted by erosion processes (Evrard et al., 2020). Consequently, the establishment of a recent age model in this context is complex. The comparison between sediment core properties recording major events (e.g. EENE, ENE and CENE) and the E index developed by Takahashi et al. (2011) was therefore one of the only methods available to date this natural archive, although this method remains perfectible. Nevertheless, the comparison of the sedimentation rates reconstructed based on this core with independent proxies for age model validation such as the volume of sediment accumulated in the reservoir reconstructed by bathymetric survey as well as sediment fluxes reconstructed from gauging stations show some consistency (Fig. 2). Reconstructed volumes of sediment accumulated during major ENE (1978-2019 periods) at the scale of the reservoir were strongly ($r^2$=0.9) correlated with reconstructed sedimentation rates of this present study (Marin, 2020). These events have induced a significant reduction of the reservoir capacity with the input of large amounts of sediment, respectively 83 million $m^3$ for the EENE 1982-1983, 75 million $m^3$ for the EENE 1997-1998, 50 million $m^3$ for the ENE 2008-2009 and 37 million $m^3$ for the CENE 2016-2017 (Marin, 2020), (Fig. 2). In addition, the annual reconstruction of sediment yield performed by Tote et al. (2011) for the Poechos reservoir covering the 1976-2005 periods compared with the thickness of annual deposits at the coring site shows some consistency ($r^2$= 0.72, the largest errors appear to be related to the periods of least input).The sedimentary sequence of the Poechos reservoir recorded two contrasted phases of sediment transfers at the coring site associated with the climate and vegetation changes during the pre and post EP-ENSO and the contemporary human activities observed along the river network. Before 1997, the sedimentation rate recorded in the reservoir remained mainly constant (5.7 cm $yr^{-1}$, SD 2.5 cm $yr^{-1}$ for the 1978-1997 period) with the exception of the EENE 1982-1983 (10 cm $yr^{-1}$), which was characterized by an acceleration of SR during this humid period (Fig. 2). The absence of coarser material layers during the EENE1982-1983 and the absence of significant change in detrital proxies (Sr, Ca/Ti) between 1978 and 1996 suggest a period of less intense hydrosedimentary dynamics (Fig. 3) at the coring site. We can also hypothesize that during this period the coring site was located further away from the deltaic area and was less sensitive to record coarser events. Source apportionment during this first phase underlines a succession of sediment supplies originating from the Andean source during the dry periods and a greater contribution of the lowland dry forest during the humid periods (Fig. 6). This source apportionment is mainly controlled by the rainfall distribution, which is in turn controlled by the prominent topography of the Andean mountains and the interaction between atmospheric moisture flux and orography (Rau et al., 2017). On the dry western flank of Peruvian Andes, mean annual rainfall is estimated to reach 10-15 mm $yr^{-1}$ below 500 m a.s.l (90 mm $yr^{-1}$ in the northern Pacific coast, dry forest area) whereas annual rainfall can exceed 1000 mm $yr^{-1}$ around 1000 and 3000 m a.s.l on the western side of the Peruvians Andes (Arias et al., 2021). ENE, EENE and CENE associated with warm temperatures along the Peruvian coast (positive E index) disrupt this rainfall pattern with heavy precipitation occurring on the western dry flank of Peruvian Andes along the lowland areas occupied by the dry forest characterized by a low vegetation cover prone to erosion. Rainfall in Piura City, located to the south of the Poechos reservoir amounts to 60 mm $yr^{-1}$ during a

normal year whereas during the EENE of 1982-1983 and 1997-1998, annual rainfall of 2150 and 1800 mm yr$^{-1}$ was recorded (Takahashi, 2004). This climatic specificity influenced by ENSO therefore explains the contrasted source contributions observed in the Poechos reservoir during the warm and cold phases.

A significant acceleration of sediment delivery was observed during the 1990s with the occurrence of a tipping point around the EENE 1997-1998. From this period onwards, sedimentation rate increased by 140% (Fig. 2 & 6). Previous studies have demonstrated that the post EP-ENSO dynamics are significantly distinct from the previous dynamics in the Catamayo-Chira catchment with a significant decrease in sediment flux during post EP-ENSO periods (Tote et al., 2011). According to these authors, after an EENE, the sediment stored in alluvial plains of the lowest part of the catchment are eliminated due to channel

enlargement and deepening. In post EP-ENSO periods, the normal discharges and persistent sediment supplies from the middle and upper catchment parts lead to river aggradation and to the storage of substantial amounts of sediment in alluvial plains. The decline in sediment export is likely to persist for several years until equilibrium is restored. This observation has also been highlighted on a larger spatial (western Andes) and a longer temporal scale by Morera et al. (2017) by analysing unpublished dataset of suspended sediment yields covering the 1968-2012 period. During EENE, suspended sediment yield increased by

3–60 times as a result of a sharp increase in river water discharge due to high precipitation rates and the associated increase in transport capacity. As reported by Tote et al. (2011), sediments accumulate in the mountain and piedmont areas during dry normal years, and are then rapidly mobilized during the EENE or CENE.

The absence of strong EP-ENSO event for more than 14 years (between 1983 and 1997) may have promoted the storage of a significant amount of sediment in the alluvial plains upstream of the Poechos reservoir. This large volume of sediment was

mobilized and exported by the EENE 1997-1998 resulting in the highest sedimentation rates recorded in this sedimentary sequence. It can be assumed that this event of 1997-1998 caused significant aggradation of the deltaic zone of the Poechos reservoir in which the core was collected nearby. The increase in sedimentation rates at the core site after this event may be caused by a better connection between the core site and the alluvial plain as well as a greater succession of EP-ENSO events during the 1997-2019 period in comparison to the 1982-1997 period, mostly dry. This result demonstrated the consequence of

the pre and post EP-ENSO alternation in the progressive and rapid filling of the reservoir at the scale of 20 years as supported by previous research (e.g. Marin, 2020; Tote et al., 2011). The greater connectivity (channel enlargement, human managements) and the aggradation of the deltaic area with the succession of EP-ENSO events increased the deposition of coarser particles in the reservoir, which is reflected by the change in particle size observed in the investigated core. These coarser layers were associated with the occurrence of major EENE, ENE and CENE (Fig. 5). Before 1997, the ENE were not

specifically associated with the deposition of coarser particles. Of note, the thickness of these layers increased by 220% (moving from a thickness of 7.5 to 24 cm) from the EENE 1997-1998 (first layer observed) to the CENE 2016-2017 (last extreme event layer detected) while the average grain size increased by 190% between both events probably in response to the progressive aggradation of the delta at the coring site.

Other factors or combination of factors can also explain an acceleration of the sediment dynamics since 1997. The Poechos

reservoir is surrounded by rivers mainly located in Peru (dry forest biome), with low baseflow of +-10 L/s during the dry

season. The occurrence of a low but permanent flow in an arid and semiarid region makes agricultural practices possible. For this reason, the inhabitants, mostly farmers, took possession of the floodplain for agriculture development. Each section and stretch of the river was assigned an owner, taking charge of delineating his property every year when the level of the river allows it. Depending on rainfall and hydrological characteristics (El Niño or La Niña events and their respective intensity) and

associated sediment transport, sediment quantity available in the floodplain may be insufficient to conduct farming especially after the EENE or the CENE where channels were enlarging and deepening. In this situation, farmers used heavy machinery to transport soil and sediment material from nearby rivers or hillsides in order to provide sufficient substrate for planting crops requiring a short vegetative period (e.g. vegetables). The construction of the Poechos reservoir generated a wider water channel which was used for agriculture, with crops planted directly on the river bed with some land filling on the slopes to create an

optimal area for the farming activity. In this way, year after year, the population substantially increased the availability of sediments in the main channel of the rivers. In addition, human activity is responsible for making large volumes of sediment available for transport to the Poechos reservoir during the first floods of the wet periods. Due to the stationarity of rainfall, the process is repeated annually, progressively raising the topographic level of the riverbed and generating more sediment transport both by river dredging and soil levelling by farmers. Succession of pre and post EP-ENSO events associated with landscape

management during post EP- ENSO after the mid of 1990s may explain the acceleration of sedimentation rates and the greater contribution of the dry forest source to the reservoir siltation. Other processes that took place in the upper basin and associated with tree logging to develop agriculture or to exploit wood (Morocho, 2004) which increased in recent years can also explain a part of these accelerated sediment dynamics. As reported by (Oñate-Valdivieso and Bosque Sendra, 2010) socio-economic factors including the supply and demand of certain agricultural products, migration, economic conditions of the region,

political decisions, etc., as well as the occurrence of weather events such as droughts or floods control the expansion of cultivated areas, which complicated the establishment of the model of the evolution of land use change and particularly the extension of croplands in this area. Although the land use changes reported by Oñate-Valdivieso and Bosque Sendra (2010) are modest and do not cover the recent period (between 1986 and 2001), it demonstrates that land use changes were occurring in the dry forest area (in the middle and upper parts of the catchment). However, for dry forests, until 2014 in the Lancones

district, 50.6% of the vegetation cover was in different state of degradation processes caused by anthropogenic factors (Otivo M. et al., 2014). Understanding in detail the processes of change in coverage in this catchment remains a challenge (availability of images without cloud cover). Further studies will therefore be required to differentiate the role of climate and its consequences on vegetation cover and human activities.

The Poechos reservoir provides a representative illustration on how water bodies may be impacted by widespread changes in

land covers associated with alternation of dry and humid periods and extreme climatic events. Marin, (2020) demonstrated that in 41 years (1976-2017), the Poechos dam lost 58.5% of its original storage capacity and that just two major events have contributed to 49% of the dam siltation observed between 1976 and 2005 (Tote et al., 2011). In addition, Rosas et al. (2020) have demonstrated the vulnerability of Andean hydroelectric reservoirs against future climate change by modelling the storage capacity of the Capillucas reservoir. Ten scenarios were created using different precipitation patterns and rates, and it was

found that the average sediment load of the Cañete River was 981 kTon/yr upstream of the reservoir. Based on these scenarios, the calculated life span of the reservoir ranged from 7 to 31 years, with even the most optimistic scenario falling short of the expected 50-year functionality. Changes in precipitation patterns resulting from global climate change can have negative consequences on Andean storage reservoirs. One of the main issues is the risk of increased siltation, which could potentially compromise the access to water resources provided by the reservoirs, thereby affecting the local communities.

The case of the Poechos catchment and more generally the Andean coast is not unique although exemplative of large scale changes in precipitation patterns, post EP-ENSO vegetation dynamics and human activities along the riverine network (13,565 km²), given the immense volume of sediment transported compared to the size of the reservoir storage capacity ($1000\times10^6$m³). Other examples of severe dam siltation were observed all around the world as in Africa or in India although in general in smaller reservoirs (e.g. storage capacity of $17\times10^6$m³) (e.g. Adongo et al., 2014). This study therefore demonstrates the

vulnerability of the Andean region in comparison to other regions of the world to climate change, vegetation response to these changes and the alarming ongoing land use changes in South-America (Zalles et al., 2021). The projected population growth for this region, already one the most densely populated mountain regions in the world and the stagnating productivity will promote the expansion of agriculture at the expense of native grasslands-wetlands and forests ecosystems (Velez et al., 2021) and will increase the pressure on soil and water resources .The identification of sediment sources and their relationship with

sediment dynamics can improve the implementation of soil conservation measures of upper basin areas to limit the adverse effects of accelerated erosion on reservoir siltation through the design of efficient landscape management strategies (Vanacker et al., 2022).

## 5 Conclusions

This study highlights the potential of sediment cores for retrospectively reconstructing the interactions between El Niño-
Southern Oscillation, vegetation cover changes and agricultural practices on the siltation of reservoirs, with the example of the Poechos Reservoir, Northern Peru (1978-2019). The current study demonstrated the strong relationship between the humid and dry phases of ENSO with the mobilization of different sources of sediment. The occurrence of stationary rainfall during the ENE, EENE and CENE increased the sediment contribution of the dry forest source, a biome characterized by a poor vegetation cover of the soil, which is therefore particularly exposed to erosion during extreme rainfall events. In post EP-
ENSO periods, the normal discharges and persistent sediment supplies from the middle and upper catchment parts lead to river aggradation and the storage of substantial amounts of sediment in alluvial plains. The absence of strong EP-ENSO event between 1982 and 1997 associated with the development of agriculture along the river system of the dry forest system have led to river aggradation and the storage of substantial amounts of sediment in alluvial plains of the lowest part of the catchment. This large volume of sediment was exported during the EENE 1997-1998 inducing the aggradation of the delta at the coring
site and a substantial decrease of the storage capacity of the reservoir.

As climate changes and agriculture expansion are both expected to continue to increase in the coming decades in South-America generally and in Peru in particular, the lifespan of this reservoir is threatened over the short term as well as the whole of the socio-economic activities which depend on it in one of the most densely populated mountain regions on earth. It is therefore crucial to improve land management and soil conservation across the basin to contribute to the slowing-down of the sediment transfers in this fragile environment and to protect the soil and water resources.

## Author contribution

J.M, M.S and J.O have collected the sediment core. A.F has performed the analyses described in this manuscript. A.F, O.E, J.M, M.S and J.O have participated to the results discussion and drafted the main manuscript.

## Competing interest

The authors declare that they have no conflict of interest

## Acknowledgements

The authors are grateful to the DOSEO platform (Université Paris-Saclay, CEA, List) and especially to Anne-Catherine Simon and Mathieu Agelou for performing the tomography scanner measurements on the core sections. This work was supported by ANR PIA (funding ANR-20-IDEES-0002) and by ProCiencia (https://prociencia.gob.pe) through the project "Implementación del sistema de Monitoreo de los Sedimentos Ante los Riesgos y Desastres", contract N°011-2018-FONDECYT/BM-Mejoramiento de infraestructura para la investigación, as well as the FONDES project through the "Fortalecimiento de la red de monitoreo de sedimentos y la sedimentación en los embalses de Poechos y Gallito Ciego" and the Technical Cooperation Project RLA-5076: "Strengthening Surveillance Systems and Monitoring Programmes of Hydraulic Facilities Using Nuclear Techniques to Assess Sedimentation Impacts as Environmental and Social Risks". It inspired the AVATAR project (ANR-22-CE93-0001) funded by ANR (France) and SNF (Switzerland). In addition, we would like to thank the Proyecto Especial Chira Piura for having provided valuable information for our study. Finally, we would like to thank the two anonymous reviewers for their thoughtful comments and efforts that helped improving our manuscript.

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

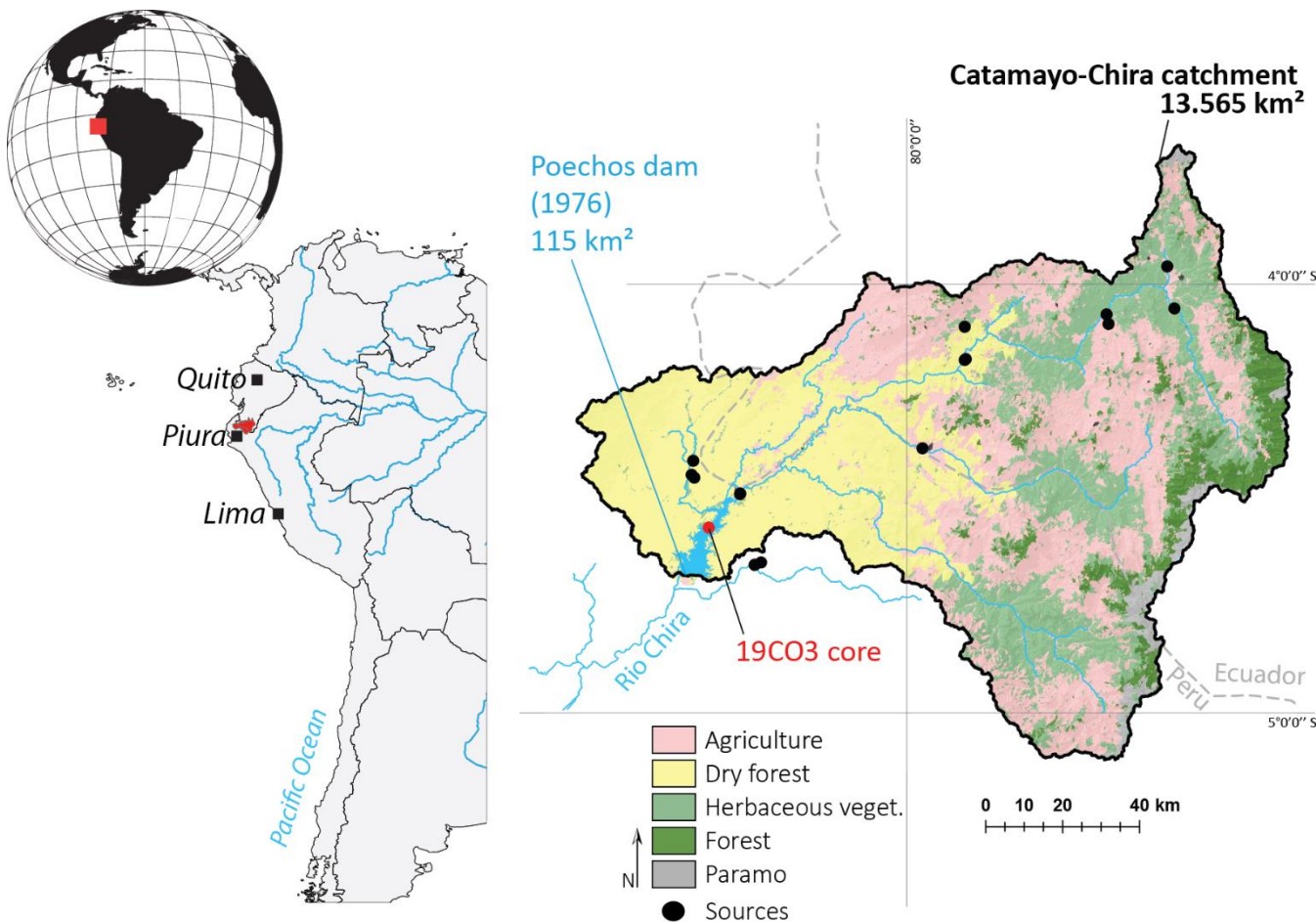

**Fig. 1 Left- Regional location of the Catamayo-Chira Basin along the western Andes at the boundary between Ecuador and Peru. Right – general setting of the Catamayo-Chira Basin with the simplified land cover 2016 provided by this study. The dry forest class corresponds to the lowland dry forest source whereas the Andean mountains source corresponds to the forest, herbaceous vegetation and agriculture classes.**

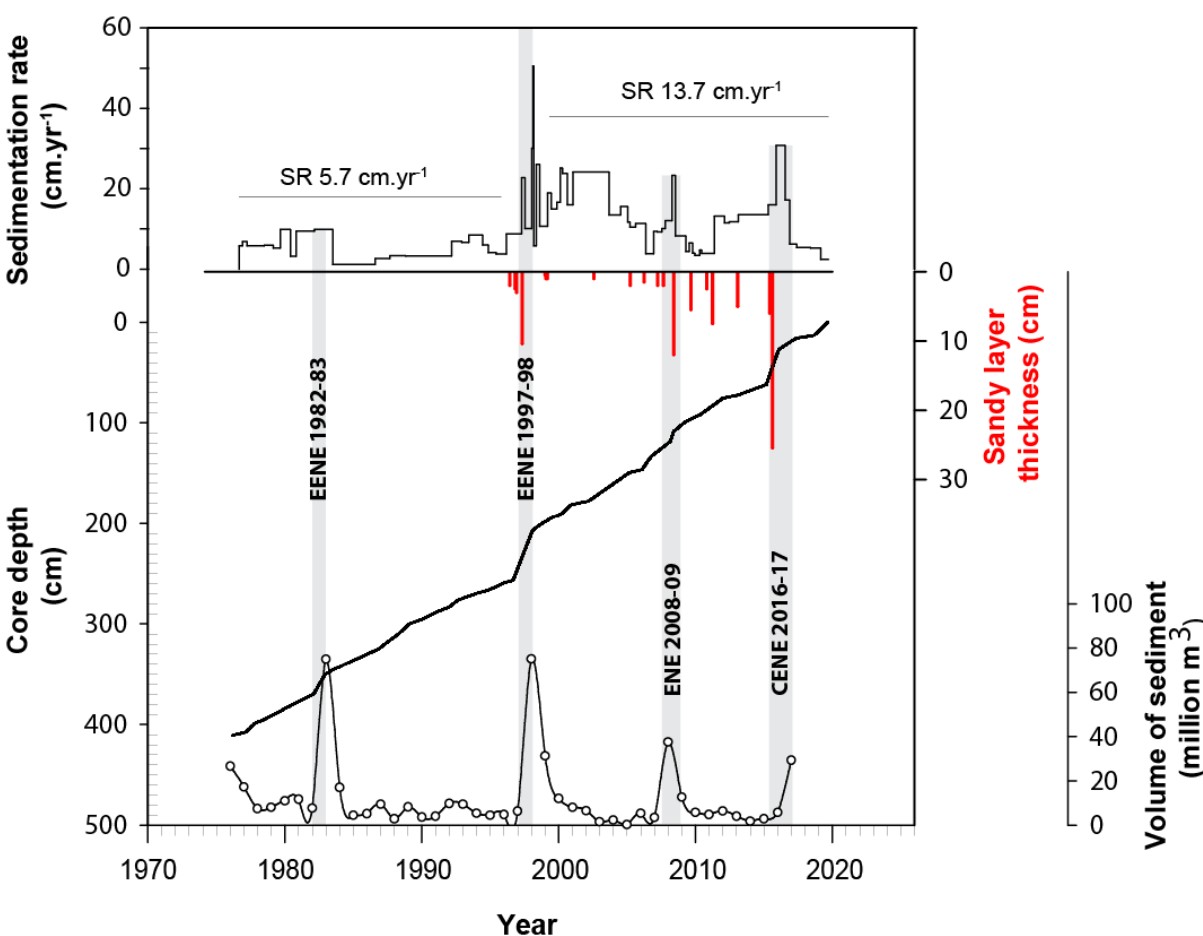


**Fig. 2 Core chronology of the 19C03 core established after correlation between the CT-number and the Eastern Pacific Index (SR=sedimentation rate). Volume of sediment from** Marin (2020)**.**

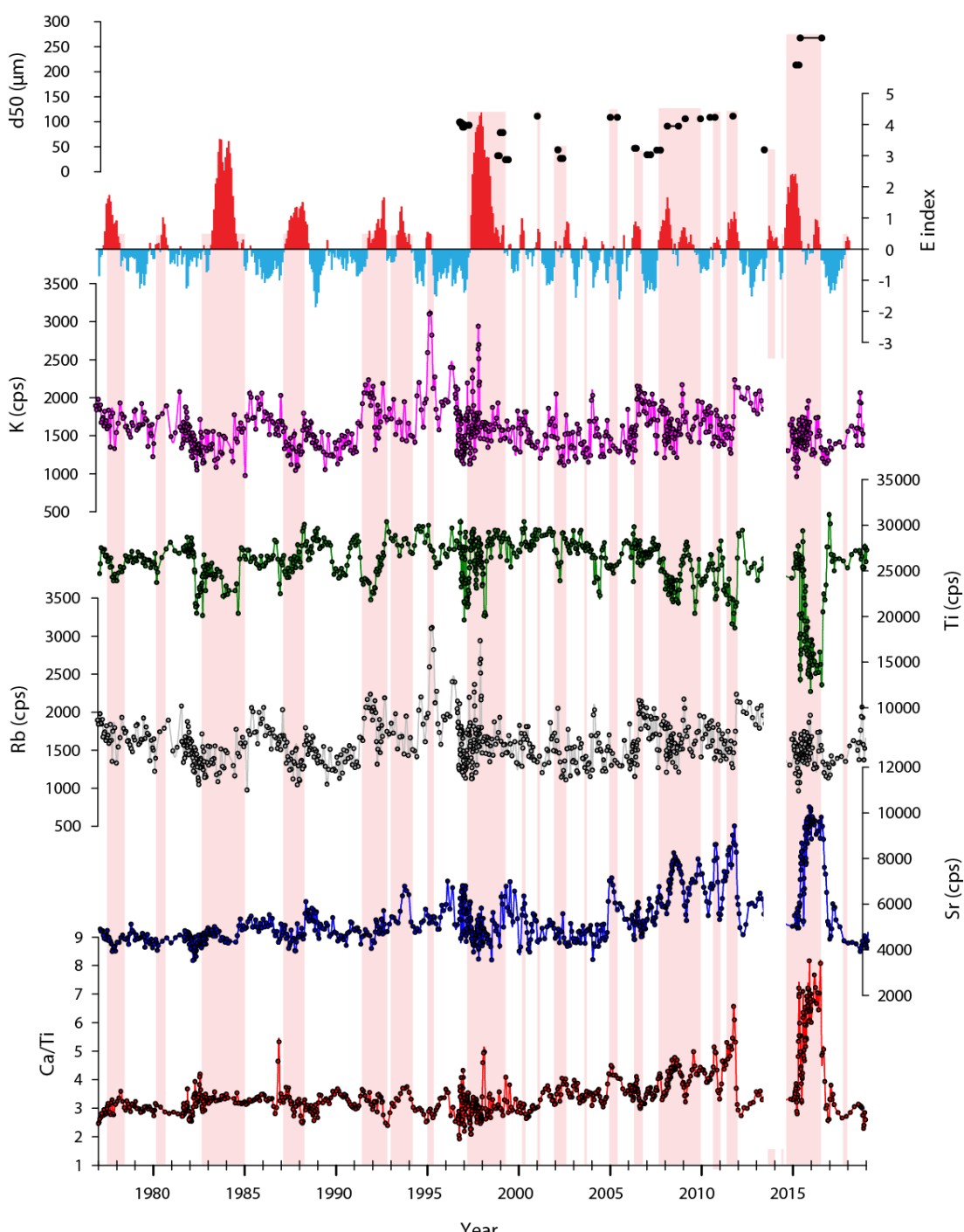

    Fig. **3** Evolution of sediment properties (chemical elements analysed with an XRF core scanner) along the 19C03 core. E

index (East Pacific Index) is freely downloadable at http://190.187.237.251/datos/ecindex_ersstv5.txt.

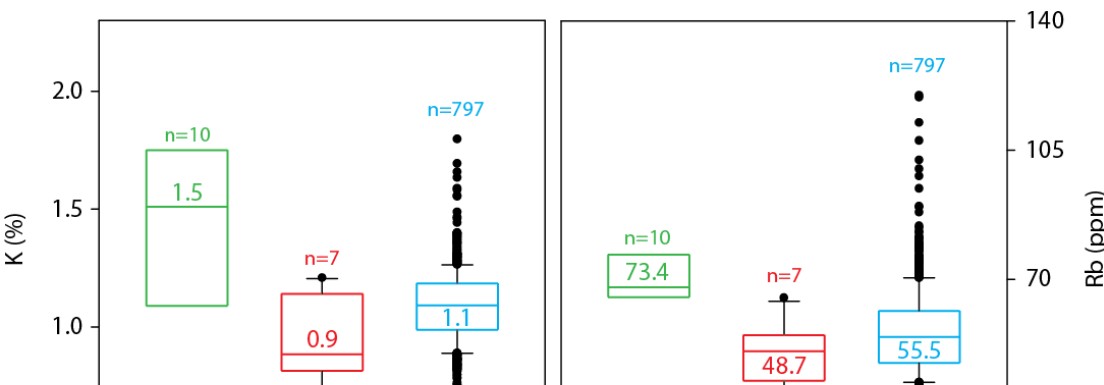

**Fig 4 Boxplot of the selected tracer properties (K and Rb) in both the sediment target samples and the potential sources (S1 and S2 correspond respectively to the Andean mountains and the Lowland dry forest sources)**


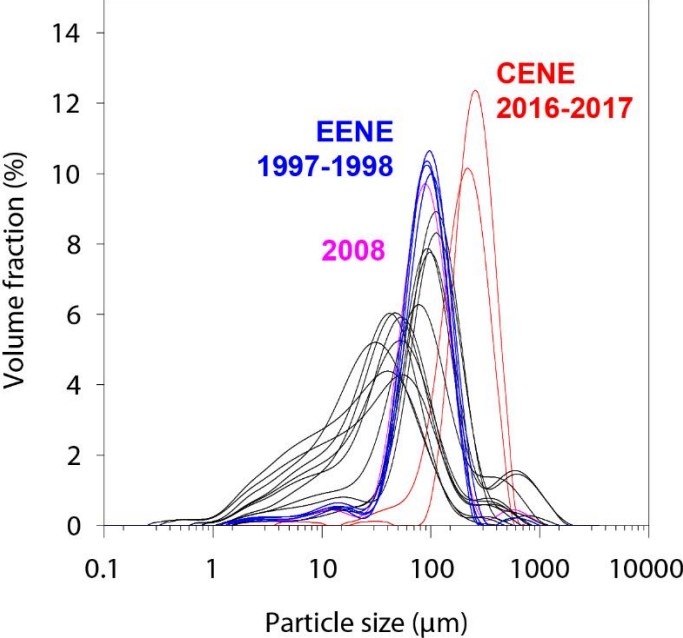

**Fig. 5 Particle size distributions of the 19 coarser layers detected in the 19CO3 core (EENE: Extreme El Niño Event, CENE: Coastal El Niño Event)**


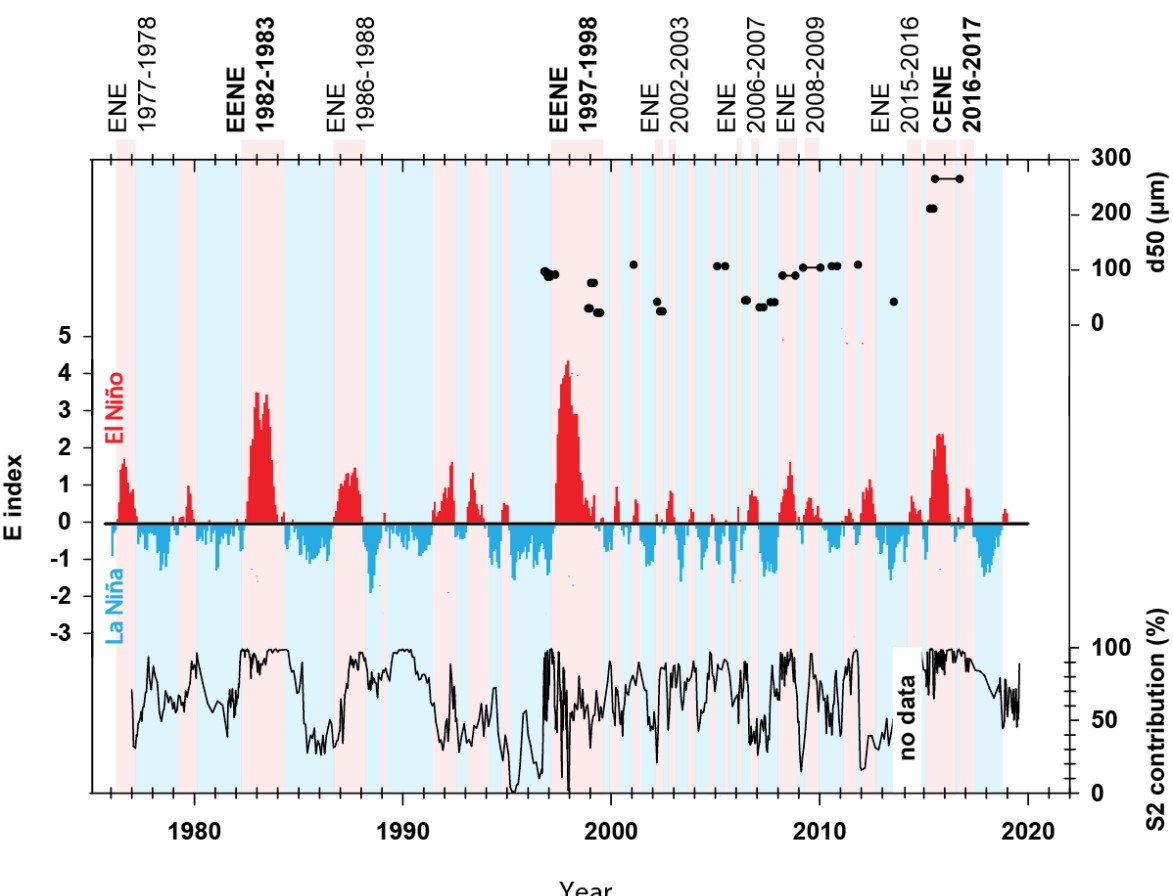

Fig 6. Comparison between sediment sources (S2 contribution = lowland dry forest source), E index (East Pacific Index: downloadable at http://190.187.237.251/datos/ecindex_ersstv5.txt) and the 19 coarser sediment layers identified along the 19CO3 core.

**Table 1 Properties of the 19 coarser layers identified in the 19CO3 sediment core**

| layer ID | min Depth (cm) | max Depth (cm) | Thickness (cm) | Min Age | Max Age | d10 | SD | d50 | SD | d90 | SD |
|---|---|---|---|---|---|---|---|---|---|---|---|
| 1 | 21 | 46.5 | 25.5 | 2016.7 | 2015.5 | 158 | 0.2 | 267 | 0.5 | 439 | 1.8 |
| 2 | 49.5 | 55.5 | 6 | 2015.4 | 2015.3 | 99 | 0.3 | 213 | 0.5 | 381 | 1.1 |
| 3 | 70 | 75 | 5 | 2013.5 | 2002.2 | 10 | 0.2 | 44 | 0.6 | 131 | 5.4 |
| 4 | 77.5 | 85 | 7.5 | 2011.8 | 2001.0 | 46 | 0.5 | 111 | 0.4 | 225 | 1.8 |
| 5 | 87.5 | 90 | 2.5 | 2010.8 | 2010.5 | 41 | 1.2 | 109 | 2.1 | 490 | 84.0 |
| 6 | 94 | 99.5 | 5.5 | 2010.0 | 2009.2 | 43 | 1.0 | 106 | 2.0 | 521 | 77.3 |
| 7 | 103.5 | 11.,5 | 12 | 2008.8 | 2008.2 | 43 | 0.1 | 91 | 0.2 | 178 | 1.6 |
| 8 | 122 | 124 | 2 | 2007.8 | 2007.6 | 6 | 0.1 | 43 | 0.3 | 144 | 3.1 |
| 9 | 128 | 130 | 2 | 2007.3 | 2007.1 | 6 | 0 | 34 | 0.1 | 91 | 0.4 |
| 10 | 139 | 140,5 | 1.5 | 2006.5 | 2006.4 | 7 | 0.1 | 47 | 0.5 | 144 | 4.8 |
| 11 | 149 | 151 | 2 | 2005.4 | 2005.1 | 19 | 0.6 | 109 | 0.4 | 234 | 2.4 |
| 12 | 172.5 | 173.5 | 1 | 2002.4 | 2002.3 | 5 | 0 | 27 | 0.2 | 83 | 0.9 |
| 13 | 195 | 196 | 1 | 1999.4 | 1999.3 | 3 | 0 | 24 | 0.1 | 87 | 0.6 |
| 14 | 197 | 198 | 1 | 1999.1 | 1999.0 | 14 | 0.5 | 78 | 1.2 | 300 | 29.8 |
| 15 | 198.5 | 199 | 0.5 | 1998.9 | 1998.9 | 4 | 0.1 | 32 | 1.0 | 126 | 3.2 |
| 16 | 234.5 | 242 | 7.5 | 1997.3 | 1997.1 | 44 | 0.3 | 93 | 0.1 | 169 | 0.6 |
| 17 | 244 | 247 | 3 | 1997.0 | 1996.9 | 36 | 0.6 | 89 | 0.4 | 159 | 1.1 |
| 18 | 248 | 250.5 | 2.5 | 1996.9 | 1996.8 | 47 | 0.2 | 97 | 0.2 | 169 | 0.9 |
| 19 | 251 | 253 | 2 | 1996.8 | 1996.8 | 42 | 0.2 | 99 | 0.1 | 181 | 0.3 |
