# Peer review of "ENSO-driven hypersedimentation in the Poechos reservoir, northern Peru"

_EGUsphere, 2022_

## Author Comment (AC1)

We would like to thank the anonymous referee for his/her careful review of the manuscript and for providing these comments and suggestions to which we respond in detail below.

| Reviewer's comment | Reply |
|---|---|
| Over the past two decades, there has been an impressive amount of research on ENSO activity in the Eastern Pacific and its impact on precipitation in the arid western coast of South America. These studies by e.g. Takahashi and Martinez (2019), Carréric et al. (2020) have related ENSO regimes with SST anomalies in the tropical Pacific Ocean, and proposed two new indices to describe the ENSO regimes: the "E and C index". Given the strong focus of the study by Foucher et al. on sedimentation rates-sources and ENSO regimes, a revision of the introduction is necessary to account for recent findings on ENSO events. The use of one index (i.e. "E index") instead of the two indices ("E and C") merits to be clarified and eventually revised. When revising the manuscript, it is recommended to use internationally agreed abbreviations for specific ENSO events like "extreme El Niño events" or "eastern Pacific ENSO" instead of introducing new abbreviations like EENE, CENE (L55-60). | If we have the opportunity to revise this manuscript, we will detail the use of E and C indices in the introduction and we will homogenize the document in order to meet the international standards.
In this manuscript, it was likely more appropriate to focus on the C index, which is strongly correlated with precipitation in the Piura-Catamayo region. This choice will be further detailed in the revised version. |
| The arid western coast of Peru has been home to agriculture-based societies for several millennia, and they have profoundly modified the landscape. There exist several studies on legacy sediments, for example, in the Chicama Valley that showed how farmers adapted the local environment through e.g. irrigation and farming infrastructure (see e.g. Caramanica, 2022). In the study, Foucher et al. highlight "management phase…soil disturbance which may exacerbate the transport of sediment to lower river sections… (L65-66)", but it is not clear if they refer to recent farming activities or also account for legacy of historical occupations. | Although historical management may have consequences on current sediment connectivity, we refer mainly to contemporary management modes in the current research. These recent mechanization developments and modern farming operations are responsible for the accelerated sediment transfer and the increased connectivity observed in the current research. Nevertheless, we found the Reviewer comment very valuable and we will introduce this concept of historical occupation when revising the introduction. |
| In the manuscript, the authors refer to the "sedimentary cascade", "sediment sources", 'sediment dynamics', "soil and water resources' and 'accelerated soil erosion". The authors intermix these terms in the introduction, without clear demarcation of the study. It is therefore not clear if they will "…estimating…sedimentation rates…sediment sources…" (L82-83) or if they will analyse "…the sedimentary cascade… (L85). The study by | We thank the Reviewer for pointing out the comprehension difficulties related to the use of these different terms. We will take this remark into account to homogenize the terminology if we have the opportunity to submit a revised version of the manuscript. |

| | |
|---|---|
| Mettier et al. (2009) on sediment sources in the region might be useful, as it contains several illustrations of the channel systems. | |
| the Catamayo-Chira basins are probably amongst the ones that are most studied in the region. The current description of the study area is very much focused on the land cover map of 2016, and some qualitative statements on recent deforestation. Please have a look at Oñate-Valdivieso (2010) for a quantitative assessment of land cover change, Arteaga-Marín (2022) for soil erosion estimates and Morera et al. (2017) and Rosas et al. (2023) for an overview of spatial variation in sediment yields along western Andes. | We thank the Reviewer for these suggestions of additional references. We will incorporate them in the revised version of the manuscript. |
| This concerns the sampling procedure. For example, it is not entirely clear where the sediment core was taken (with respect to the sediment body in the reservoir), and how representative the core was for deriving reservoir sedimentation rates. What about reworking/remobilisation of sediments in the reservoir? Also, the source sampling is not clearly described. It is unclear where the samples were taken with respect to geology, land cover, and topography. Also, why did the authors target soil samples when the material that is transported in the stream is also sourced from deeper via deep-seated landsliding, bank erosion and gullies? | The core was collected in the deltaic part of the reservoir (for technical reasons) but also to record more directly the deposits associated with floods. We do not extrapolate these results to the entire reservoir but we use instead the bathymetric data collected annually to estimate the filling rates of the reservoir. These specific aspects will be further explained in the revised version of the manuscript.
 For the sampling of sediment sources we have not given sufficient detail, as also pointed out by Reviewer 2. We will add this information in the revised version.
 Indeed, the source samples were collected according to the geology of the catchment (two main units), which itself controls the vegetation. Dry forest develops on sedimentary rocks while wet forest/grasslands/agriculture are found on the Andean volcanic rocks. The aim here was not to distinguish the surface and sub-surface sources of sediment but instead to identify the main geological source. Samples were collected in areas exposed to surface erosion without visible landside or deep gully erosion process. |
| The rationale behind the establishment of the core chronology is not clear. By directly correlating core characteristics with ENSO variability for the age-depth model, the study already imposes a relationship between sediment characteristics and ENSO events, and hence sedimentation rates. The FRN data are not helpful as independent control, but the uncertainty on the age-depth model should be reported and accordingly discussed as this has an impact on the following results. | Since this sedimentary record is depleted in fallout radionuclides (such as potential sediment source samples; the measured values will be provided in the revised version), we had to find an alternative method to date this core. We understand the reviewer's concern about the occurrence of a potential bias in the dating and will discuss this further into details in the revised version.

 Although this age model is not perfect, we were able to validate it through the identification of |

| | sediment deposits associated with extreme events since 1982. The periods of high sediment accumulation in the core correspond to those periods identified by the bathymetric surveys, an independent method of validation. |
|---|---|
| the variability in sedimentation rates and sediment sourcing are interpreted in terms of climate and land use change. The link between climate variability and sedimentation rates and sources is somehow difficult to assess in the current version of the manuscript because of the lack of an independent age control on the core. Therefore, an uncertainty analysis might be useful. Also, land use is cited to be triggering sediment transport in the lower part of the basin, mainly as a result of agricultural activities and deforestation. It would be useful to link these observations with land use change maps or data, to verify the extent and the location of the land use changes. Although previous studies have shown how land use can accelerate soil erosion in the tropical Andes, it is not yet clear how this impact is noticeable at larger spatial scales (see e.g. Vanacker et al., 2022 or Tote et al., 2011). | We believe that a way to independently validate the age model is to compare the sediment core results with the bathymetric data (Fig. 2). The EENE or ENE events, especially those of 1982, 1997 and 2016, are well correlated both in the sedimentary archive and in the volume of sediment accumulated in the reservoir, estimated by bathymetric surveys.

 We thank the reviewer for his advice to add a map or data to support our message on land cover changes. This is a very good suggestion that we will take it into account in the revision of the manuscript. |
| Abstract: The abstract contains a number of specific terms that would need to be introduced and defined beforehand. This concerns – for example – the definition of « extreme el nino events » or « coastal el nino events » or « C and E index ». Also, please check if the use of abbreviations is necessary, and not overloading the text.

 Please check the language and writing style of the document, particularly the use of capitals for nouns like « Volcanic (L97) », « Economically Active Population (L103) » etc. | We thank the reviewer for this valuable comment. We will take it into account when revising the manuscript. |
| L1-3 : title sounds very dramatic « … threaten soil and water resources through hyper sedimentation ». Can you rephrase into a more objective statement, for example, indicating how much sedimentation rates increased during these events ? | This title is indeed dramatic, but in our opinion, it describes the situation quite well.
 Soil erosion is so extensive in the study area that soils/sediment are depleted in fallout radionuclides (which leads to problems for drawing an independent age model as the reviewer pointed out).
 After each El Niño event, farmers have to move thousands of m3 of soil to be able to cultivate again in the vicinity the river systems (the soil being transported away to the reservoir after flood events). |

| | The reservoir, one of the largest in the region, is vital for flood control during El Niño events, irrigation and for supplying food commodities, and this is threatened by this hyper-sedimentation (at the scale of the next two decades). If the Reviewer and Editor agree, we would like to keep this title. |
|---|---|
| Technical corrections | We thank the reviewer for highlighting multiple points for improvement in the manuscript. We find the comments in the "technical correction" section justified. We will take them individually into account if we have the opportunity to submit a revised version of our manuscript. |

---

## Author Comment (AC2)

We would like to thank the anonymous referee for his/her careful review of the manuscript and for providing these comments and suggestions to which we respond in detail below.

| Reviewer's comment | Reply |
|---|---|
| I suggest the authors include some brief descriptions of Extreme El Niño Events and Coastal El Niño Events to have a complete idea of their differences or impacts in the process studied. | To facilitate the understanding of these phenomena on sediment dynamics, we will describe the characteristics of these events if we have the opportunity to revise this manuscript. |
| Sampling: how many samples of each source were collected; how many subsamples composed the sample? | We forgot to specify these technical elements. In this study we analyzed 13 composite samples composed of 5 subsamples. We will add this information in the revised version. |
| Considering laboratory analysis, why did the authors choose the chemical elements described as tracing properties (Ti, K, Sr, Rb)? | The K and Rb were selected because they statistically differentiate the two sediment sources (Andean mountains and Lowland dry forests). We detail these results in section 3.3. Finally, we added Ti and Sr because they are classically used to identify detrital inputs and particle size changes in sedimentary archives (section 2.2.2). Of note, these elements (Ti and Sr) were chosen to describe the core but were not used for the sediment tracing. This will be clarified when revising the manuscript. |
| Sediment core dating: is this the first work which uses the relationship between E index temporal series and CT data to date a sediment core? Is the coefficient of determination obtained (0.45) acceptable for these studies? Is there a statistical significance value reported in this analysis? | The dating of this core was challenging since sediment was depleted in fallout radionuclides in this region (erosion rates are particularly high and initial radionuclide deposition was limited in equatorial regions). This approach of correlating climate data with other data measured in sediment cores for estimating age model is not new. We will give examples of previous research using this technique in the revised version. In this study the $r^2$ is not high mainly because we did not compare rainfall data with sediment fluxes but we used instead a less accurate monthly rainfall index. We therefore miss a certain number of rainfall events that are recorded in the core and not with this index Nevertheless, despite this $R^2$ of 0.45, we are able to identify the major El Niño events, which allowed us to validate the age model. |
| Lithology, lines 190-191: do the authors mention those main four coarse layers in the core (which coincide with the thicker) in relation to the low standard deviation value of d10, d50 and d90? | Thank you for pointing out this limitation. We will do so in the revised version. |
| Sediment sources, lines 204-209: are K and Rb contents reported calibrated values, as the ones mentioned in line 210 for the sediment core? | We mention here the calibrated values in the soils (lines 203-209) and in line 210 the values measured in the core. We will add information |

| | to avoid any ambiguity in the revised version of the manuscript. |
|---|---|
| Technical corrections | We thank the reviewer for pointing out these minor technical problems. We will address these points when revising this manuscript if we are allowed to do so. |

---

## Author Response (AR1)

**Referee 1.**

We would like to thank the anonymous referee for his/her careful review of the manuscript and for providing these comments and suggestions to which we respond in detail below.

| Reviewer's comment | Reply |
|---|---|
| Over the past two decades, there has been an impressive amount of research on ENSO activity in the Eastern Pacific and its impact on precipitation in the arid western coast of South America. These studies by e.g. Takahashi and Martinez (2019), Carréric et al. (2020) have related ENSO regimes with SST anomalies in the tropical Pacific Ocean, and proposed two new indices to describe the ENSO regimes: the "E and C index". Given the strong focus of the study by Foucher et al. on sedimentation rates-sources and ENSO regimes, a revision of the introduction is necessary to account for recent findings on ENSO events. The use of one index (i.e. "E index") instead of the two indices ("E and C") merits to be clarified and eventually revised. When revising the manuscript, it is recommended to use internationally agreed abbreviations for specific ENSO events like "extreme El Niño events" or "eastern Pacific ENSO" instead of introducing new abbreviations like EENE, CENE (L55-60). | We updated the reference about ENSO, adding Cai et al, 2021 and Geng et al 2022. They study the ENSO sea surface temperature (SST), focused on the eastern Pacific (EP) or Niño 1+2, referred to as EP-ENSO regime, which show strong warm SST anomalies in the EP. As well as, scientific publications, which study EP-ENSO anomalies as 1982-83, 1997-98 and the coastal El Niño event (2017). **Lines 70-83**

We regret that we do not explain into detailed the differences between ENSO and EENE. El Niño and La Niña are the oceanic components of ENSO. The updated version considers international acronyms, as EP or EP-ENSO, both refer to strong warm SST anomalies in the EP, or warm phase (El Niño). However, there is not international agreed abbreviations for specific evets like El Niño 1982-83, El Niño 1997-98 and 2017 the Coastal El Niño Events (CENE). In our manuscript, we do not discuss about all the historical Strong El Niño Events, EP El Niño, (e.i. 1925-26, 1982-83, 1987-88, 1997-98, 2008-09, see Takahashi and Martinez 2017). We focus only on the two very strong El Niño Events (1982-83 and 1997-98), some authors call super El Niño, due to its impacts in hydrology, sediment transport and sedimentation, for our study area and period. Because of the impact of the 1982-83 and 1997-98 El Niño events, in our study area, be call the Extreme El Niño Events (EENE), as well as, Morera et al 2017. |
| The arid western coast of Peru has been home to agriculture-based societies for several millennia, and they have profoundly modified the landscape. There exist several studies on legacy sediments, for example, in the Chicama Valley that showed how farmers adapted the local environment through e.g. irrigation and farming infrastructure (see e.g. Caramanica, 2022). In the study, Foucher et al. highlight "management phase…soil disturbance which may exacerbate the transport of sediment to lower river sections… (L65-66)", but it is not clear if they refer to recent farming activities or also account for legacy of historical occupations. | Although historical management may have consequences on current sediment connectivity, we refer mainly to contemporary management modes in the current research. These recent mechanization developments and modern farming operations are responsible for the accelerated sediment transfer and the increased connectivity observed in the current research. Nevertheless, we found the Reviewer comment very valuable and we have introduce this concept of historical occupation **Lines 91-93.** |

| | |
|---|---|
| In the manuscript, the authors refer to the "sedimentary cascade", "sediment sources", 'sediment dynamics', "soil and water resources' and 'accelerated soil erosion". The authors intermix these terms in the introduction, without clear demarcation of the study. It is therefore not clear if they will "…estimating…sedimentation rates…sediment sources…" (L82-83) or if they will analyse "…the sedimentary cascade… (L85). The study by Mettier et al. (2009) on sediment sources in the region might be useful, as it contains several illustrations of the channel systems. | We thank the Reviewer for pointing out the comprehension difficulties related to the use of these different terms. The terminology was homogenized in this revised version of the manuscript. |
| the Catamayo-Chira basins are probably amongst the ones that are most studied in the region. The current description of the study area is very much focused on the land cover map of 2016, and some qualitative statements on recent deforestation. Please have a look at Oñate-Valdivieso (2010) for a quantitative assessment of land cover change, Arteaga-Marín (2022) for soil erosion estimates and Morera et al. (2017) and Rosas et al. (2023) for an overview of spatial variation in sediment yields along western Andes. | We thank the Reviewer for these suggestions of additional references. These references have been added to the manuscript (**e.g. Lines 84-86, 142-145, 373-382).** |
| This concerns the sampling procedure. For example, it is not entirely clear where the sediment core was taken (with respect to the sediment body in the reservoir), and how representative the core was for deriving reservoir sedimentation rates. What about reworking/remobilisation of sediments in the reservoir? Also, the source sampling is not clearly described. It is unclear where the samples were taken with respect to geology, land cover, and topography. Also, why did the authors target soil samples when the material that is transported in the stream is also sourced from deeper via deep-seated landsliding, bank erosion and gullies? | The sampling procedure was detailed (**Lines 158-172**) and potential biases were discussed (e.g. **lines 341-344**) |
| The rationale behind the establishment of the core chronology is not clear. By directly correlating core characteristics with ENSO variability for the age-depth model, the study already imposes a relationship between sediment characteristics and ENSO events, and hence sedimentation rates. The FRN data are not helpful as independent control, but the uncertainty on the age-depth model should be reported and accordingly discussed as this has an impact on the following results. | The constraints involved in creating an age model in this context and the choice of this technique and the associated errors were discussed (**lines 291-304**) |
| the variability in sedimentation rates and sediment sourcing are interpreted in terms of climate and land use change. The link between climate variability and sedimentation rates and | We believe that a way to independently validate the age model is to compare the sediment core results with the bathymetric data (Fig. 2). The EENE or ENE events, especially those of 1982, |

| | |
|---|---|
| sources is somehow difficult to assess in the current version of the manuscript because of the lack of an independent age control on the core. Therefore, an uncertainty analysis might be useful. Also, land use is cited to be triggering sediment transport in the lower part of the basin, mainly as a result of agricultural activities and deforestation. It would be useful to link these observations with land use change maps or data, to verify the extent and the location of the land use changes. Although previous studies have shown how land use can accelerate soil erosion in the tropical Andes, it is not yet clear how this impact is noticeable at larger spatial scales (see e.g. Vanacker et al., 2022 or Tote et al., 2011). | 1997 and 2016, are well correlated both in the sedimentary archive and in the volume of sediment accumulated in the reservoir, estimated by bathymetric surveys. This point was discussed on the updated version of the manuscript (**Lines 291-304**).

We thank the reviewer for his advice to add a map or data to support our message on land cover changes. Additional informations were added **lines 372-382**. |
| L1-3 : title sounds very dramatic « … threaten soil and water resources through hyper sedimentation ». Can you rephrase into a more objective statement, for example, indicating how much sedimentation rates increased during these events ? | The title was modified according to the Reviewer and Editor comments.
In this updated version we use the title proposed by the Editor. |
| L11 : Can you be more precise here ? You state that « EENE have always impacted hydrology in South America » but is this the case everywhere in South America, or more specific for the arid western coast of South America ? And what do you mean with « hydrology », does this also include sediment transport ? | This part was rewritten. |
| L12-13 : Not clear to me what you mean with « EENE … their intensification by global warming and their association with changes in human activities and land cover ». How are EENE's associated with land cover ? Do you refer to land cover change after EENE events ? | Yes, we are referring to land cover changes after ENSO events. We have rewritten this part. (**Line 13**). |
| L14 : rephrase « freshwater originating from large dams » What is the origin of the fresh water ? Where are the sources ? | The word "freshwater" was removed. |
| L21 : Is the « dry forest biome » contributing more sediments than the andean uplands where agricultural activities were traditionally concentrated ? Can you specify the land cover of the « forest biome » ? | The term "biome" was removed and this section was rewritten to avoid ambiguity about the contribution of this source. |

| | |
|---|---|
| L43-46 : These references supporting the increase in soil erosion, muddy floods and transport of contaminants after land use change are from studies in Western Europe with a different land use legacy. There have been multiple studies in South America on this topic, also in Peru, and it would be relevant to include also references to South American studies here. | These references have been removed and replaced by studies conducted in South America and Peru (**Lines 50-52**). |
| L53 : please check writing «the North Peru » | This term was homogenized throughout the manuscript. |
| L60: please avoid using terms like "deleterious" | This term was removed. |
| L62-65: Can you support this with a reference to the scientific literature? Also, this is an area of legacy land use (see e.g. Caramanica, 2022), what about historical land use activities? | We thank the Reviewer for this comment. We have added information about the legacy of past developments to complement current developments (**Lines 91-93**). |
| L67: Can you indicate which network this study refers to? "…data available from a network…" How many data gaps exists? | These informations were added. |
| L82: Is it necessary to refer here to the "tropical forest biome"? How would sediment processes be different in this biome? | The term biome is confusing here. It was removed. |
| L86: Please check the use of 'sedimentary cascade', this refers to something very different than "sediment sources" and "sedimentation rates". | The term sedimentary cascade was removed from this new version of the manuscript. |
| L91: check writing: "Northern" but then "western" and "eastern" | Writing was homogenized. |
| L98: what is the source of the data presented here? What is the reference for the geology? And the ecoregion map? What is the reference for the land cover map? | Reference was added and methodology for generated land cover map was explained (**Lines 220-233**). |
| L107: You might want to add reference to land use change analyses that were done for the region. See e.g. Oñate-Valdivieso (2010) | Informations from the paper of Oñate-Valdivieso (2010) were added in this current version of the manuscript (**Lines 143-145 & 373-375**). |

| | |
|---|---|
| L115: When referring to the erosion and sediment problems, you might want to refer to studies on soil erosion for the region (e.g. Arteaga-Marin et al., 2022; Morera et al., 2017; Rosas et al. 2023; Tote et al., 2011) | We thank the Reviewer for these interesting papers. We have used these references on this updated version (**Lines 53-57, 325-354**) |
| L118: Please refer here to the numbers published in Morera et al. (2017) | Done (**Lines 84-86**). |
| L122: Where was the core located with respect to the entrance/outflow of the reservoir? Was the core taken on the sediment delta? Can you indicate its location on a bathymetric map of the sediment core? How representative is one core for deriving sedimentation rates of a large lake? | Location and representativeness of the core were described in the methodology (**Lines 158-171**) and discussed on the discussion section (**Lines 339-343**). |
| L156: Why is the core chronology not based on an age-depth model with the FRN? By linking the core density with the "E-index" you cannot do an independent analysis of sediment characteristics with the ENSO variability. | We detail why we opted for this choice in the methodology and discuss the limitations associated with this age model in the first part of the discussion (**Lines 291-304**) |
| L177: In how far are the sedimentation rates robust, given the uncertainty on the age-depth model? The paper could be strengthened by quantifying and reporting the uncertainty on the rates? Are these rates conform with what is reported from bathymetry? | We compared our data with previous studies (Tote et al. 2011 & Marin, 2020) to estimate the robustness and representativeness of our reconstruction (**Lines 291-304**). |
| L194: It is not entirely clear why K is such an important element for the sediment fingerprinting. The K concentrations in the soil typically vary based on soil weathering degree, and would be much higher in sediments than in weathered soil material. Same thing for Ca concentrations that could vary between soils and sediments. | This element were kept after different steps of statistical tests (conservativity, discriminant). |
| L204: The rationale behind the selection of these two "sources" for the fingerprinting of the sediments is not clear, and would need further explanation. Previous work by e.g. Tote et al. (2011) pointed to differences in sediment dynamics pre- and post-ENSO events, whereby the material was quickly transported to the reservoir during ENSO events, and sediments were accumulating in the upstream alluvial plains of the lower basin during post-ENSO | The choice of sources and processes detailed by Tote et al (2011) have been detailed in the new version of the manuscript (**Lines 325-336**). |

| | |
|---|---|
| events. What is the expected difference in geochemistry between the "dry forest" and the "upstream sources"? Why contrasting the geochemistry of the soils with sediments, when the FRN on the sediments show that the material is probably sourced from much deeper. | |
| L246-247: Are the data on the sedimentation rates robust as to identify different periods of sedimentation (with tipping points)? Can you validate your results using the information published by e.g. Tote et al. (2011)? | This point was discussed **lines 296-304 and lines 337-352.** |
| L255-258: can you comment on the sediment transport mechanisms in the area? Is the sediment transport mainly supply or transport limited? Tote et al. (2011) refer to the wide availability of sediment in the alluvial channels pre-ENSO events, and this would imply sediment transport limited systems in the lower part. Can you specify if farmers are redistributing sediments within the floodplain, or bringing sediments from elsewhere? | We thank the reviewer for this comment. In this updated version, we discuss the transfer processes discussed by Tote et al (2011). **Lines 337-352**. |
| L278: There are recent studies showing the sensitivity of reservoir sedimentation to climate variability in Peru. You might have a look at Rosas et al. (2020). | This reference statements were added to this manuscript. (**Lines 386-393**). |
| L283: The study does not demonstrate quantitatively that there is a direct impact of land use change on the sedimentation rates or sediment sources. Please reconsider this sentence | Sentences was modified. |
| L285 & following: This is a valid point and concerning for the region. You can have a look at recent work by e.g. Vanacker et al. (2022) and the references herein to under build your statement. | This reference was added. |
| Figure 1: Can you add latlong coordinates to the figures? The caption for the left figure mentions the USGS SRTM but there is no information on the elevation visible on the figure. Can you please check? The right-hand figure shows the | The figure was modified following the reviewer comment and reference was added. |

| land cover map of 2016. Can you indicate its source? | |
|---|---|
| Figure 3 & 6 : What is the source of the data on the climate: Where is the E-index coming from? And what is the source of this data? | Source was added on the figures 3 and 6. |

**Referee 2.**

| Reviewer's comment | Reply |
|---|---|
| I suggest the authors include some brief descriptions of Extreme El Niño Events and Coastal El Niño Events to have a complete idea of their differences or impacts in the process studied. | The description of Extreme El Niño Events and Coastal El Niño Events and their impact on sediment transfers were added in this updated version (**Lines 78-87**). |
| Sampling: how many samples of each source were collected; how many subsamples composed the sample? | We forgot to specify these technical elements. In this study we analyzed 13 composite samples composed of 5 subsamples. This information was added **Lines 178-179 and 173**. |
| Considering laboratory analysis, why did the authors choose the chemical elements described as tracing properties (Ti, K, Sr, Rb)? | The K and Rb were selected because they statistically differentiate the two sediment sources (Andean mountains and Lowland dry forests). We detail these results in section 3.3. Finally, we added Ti and Sr because they are classically used to identify detrital inputs and particle size changes in sedimentary archives (section 2.2.2). Of note, these elements (Ti and Sr) were chosen to describe the core but were not used for the sediment tracing. |
| Sediment core dating: is this the first work which uses the relationship between E index temporal series and CT data to date a sediment core? Is the coefficient of determination obtained (0.45) acceptable for these studies? Is there a statistical significance value reported in this analysis? | The constraints involved in creating an age model in this context and the choice of this technique and the associated errors were discussed (**lines 291-304**) |
| Sediment sources, lines 204-209: are K and Rb contents reported calibrated values, as the ones mentioned in line 210 for the sediment core? | We mention here the calibrated values in the soils (lines 203-209) and in line 210 the values measured in the core. |
| Technical corrections | We thank the reviewer for pointing out these minor technical problems. We will address these points when revising this manuscript if we are allowed to do so. |

---

## Author Response (AR2)

We would like to thank the anonymous referee for his/her careful review of the manuscript and for providing these comments and suggestions to which we respond in detail below.

| Reviewer's comment | Reply |
|---|---|
| It is not yet entirely clear to me how different these areas are in terms of land cover, climate, and geology although this might be very clear for the authors who are familiar with the region. I would find it helpful to have the two potential source areas delineated/indicated in Figure 1 (eventually by adding some text to the legend) | We thank the referee for mentioning this difficulty. Informations were added in the text (**Lines 175-176**) and in the title of figure 1. |
| L21: Eastern | This typo was modified |
| L23: Can you check writing? "this source contribution was ... controlled by stationary rainfall" (?) | This sentence was clarified (**Line 23**). |
| L32: amplified the quantity (?) -> There is no info on quality of sediments in the sentences above, so not clear what is being referred to | "quality" was removed from this sentence. |
| L33: lifespan of the reservoir (?) | We thank the referee for pointing out this typo. |
| L52: data on sediment loads from gauging stations, ... (?) | This suggestion was added to the text (**Line 52**). |
| L54: north of the western Peruvian Andes (3° - 6°S) | Modified (**Line 53**). |
| L55: for the northern Andes where sediment yields of .... were reported for rivers... Ecuadorian (?) Andes -> I think that Vanacker et al (2015) reports on the Ecuadorian Andes. | We thank the reviewer for pointing out this error. Columbian was replaced by Ecuadorian **line 55**. |
| L56: you can probably delete the values in mm/yr, and stick to the values reported in t/km²/yr for consistency. | Values in mm/yr was removed. |
| L58-59: I would remove "with agriculture as the main user" as it somewhat evident for "irrigation" | This part was deleted. |
| L60: Please check wording: "animal and human food supply" | This sentence was modified (**Line 60**). |
| L77: whereas OR meanwhile -> check wording | Meanwhile was removed from this sentence. |
| L83: Contrary to EENE (?) | Modified (**Line 83**) |
| L111: you could remove "of the dry forest area" as the sentence is a bit long and hard to read | As suggested by the referee this part was removed. |
| L144: "replace" instead of "intrude"? | Modified (**line 144**). |
| L145: "where natural vegetation is converted to grasslands and crops for ..." | We thank the reviewer for this suggestion (modified **Line 145**). |
| L174: "based on the main geological units that correspond to different land cover classes" (?) | This sentence was modified (**Line 175**). |
| L206-211: long sentence -> can you split the sentence in two parts for readability? | This sentence was spited in two parts (**lines 206-210**). |
| L220-225: Can you clarify if you produced a land use or land cover map, and check the wording in these lines? | A land cover map was produced. Wording was changed line **222**. |
| L234: followed by a vector edit (?) | Modified as suggested by the reviewer (**Line 234**) |

| | |
|---|---|
| L259: can you write "positive deviations" instead of "fluctuations", or is this not correct? | The term "positive deviations" is not correct here and was not modified in the manuscript. |
| L379: different state of degradation (?) | Modified (**Line 379**). |
| L385: demonstrated that | Modified (**Line 385**). |